# Patterns of metformin monotherapy discontinuation and reinitiation in people with type 2 diabetes mellitus in New Zealand

Simon Horsburgh[1,2]* , Katrina Sharples[1,3,4], David Barson[1,2], Jiaxu Zeng[1,2], Lianne Parkin[1,2]

1 Pharmacoepidemiology Research Network, University of Otago, Dunedin, New Zealand, 2 Department of Preventive and Social Medicine, Otago Medical School–Dunedin Campus, University of Otago, Dunedin, New Zealand, 3 Department of Medicine, Otago Medical School–Dunedin Campus, University of Otago, Dunedin, New Zealand, 4 Department of Mathematics and Statistics, University of Otago, Dunedin, New Zealand

☯ These authors contributed equally to this work.
* simon.horsburgh@otago.ac.nz

## Abstract

### Aim

To describe the patterns of discontinuation and reinitiation in new users of metformin monotherapy in New Zealand, overall and according to person- and healthcare-related factors.

### Materials and methods

We created a cohort (n = 85,066) of all patients in New Zealand with type 2 diabetes mellitus who initiated metformin monotherapy between 1 January 2006 and 30 September 2014 from the national data collections, and followed them until the earlier of their death or 31 December 2015. Discontinuation was defined as a gap in possession of metformin monotherapy of ≥90 days. We explored patterns of discontinuation and reinitiation using competing risks methods.

### Results

After 1 year of follow-up, 28% of cohort members had discontinued metformin monotherapy at least once; the corresponding figures after 2 and 5 years were 37% and 46%. The proportions who reinitiated metformin monotherapy within 1, 2, and 5 years of their first discontinuation were 23%, 49%, and 73%. Discontinuation after the first reinitiation was common (48% after 1 year). Discontinuation and reinitiation varied by age, ethnicity, and other person- and healthcare-related factors.

### Discussion

Our findings highlight the dynamic nature of metformin monotherapy use, show that substantial periods of non-use are common, and identify priority populations for interventions to facilitate adherence.

**Data Availability Statement:** The data underlying the results presented in the study are available from the following New Zealand Ministry of Health data collections: The Pharmaceutical Collection

(PHARMS), The National Minimum Dataset (Hospital Events) (NMDS), The National Health Index collection (NHI), The Mortality Collection (MORT). Data can be accessed via request from the sources listed above via the Ministry of Health (https://www.health.govt.nz/nz-health-statistics/access-and-use). The authors did not have any special access to the data used in this study. Please see our cover letter for more information.

**Funding:** This work was supported by the Health Research Council of New Zealand (HRC) and the Pharmaceutical Management Agency (PHARMAC) of New Zealand (HRC grant number 16/780). The funders had no role in study design, data collection and analysis, decision to publish, or preparation of the manuscript.

**Competing interests:** The authors have declared that no competing interests exist.

**Abbreviations:** CCI, Charlson Comorbidity Index; CI, Confidence Interval; CVD, Cardiovascular Disease; DHB, District Health Board; HR, Hazard Ratio; MELAA, Middle Eastern/Latin American/African; MPR, Medication Possession Ratio; NHI, National Health Index; NZDep, New Zealand Deprivation Index; Pharms, Pharmaceutical Collection; RR, Rate Ratio; T2DM, Type 2 Diabetes Mellitus; VDR, Virtual Diabetes Register.

# 1. Introduction

Type 2 diabetes mellitus (T2DM) is a growing threat to health globally; in 2019 there were an estimated 463 million adults living with diabetes, an estimated 90% of whom had T2DM [1]. In New Zealand, an estimated 6.4% of adults were living with diagnosed T2DM in 2019 and the prevalence was much higher among Māori (the indigenous people, 16.5% of the total population [2]), Pacific peoples, Indo-Asians, and those living in more socioeconomically deprived areas [3, 4]. Part of the increase in T2DM over time is being driven by the steady decrease in the age of T2DM onset [5], which is particularly concerning given evidence that earlier onset is associated with more T2DM complications and poorer outcomes [6].

Metformin monotherapy is the first-line pharmacological treatment for T2DM in New Zealand [7, 8] and accounts for approximately 85% of the initial pharmacological agents prescribed for T2DM [9]. In a previous paper, we reported that there were important differences in long-term adherence to metformin monotherapy (measured using the medication possession ratio (MPR)) among population groups in New Zealand, with Māori, Pacific, and younger people with T2DM having lower adherence [10]. While the MPR provides a useful population-level indicator for identifying differences in adherence between population groups, an important limitation is that it does not adequately address the dynamic nature of medication use. For example, it does not readily distinguish between people who continue to take metformin sporadically over a defined period of time and those who have repeated extended gaps in metformin possession. These are two quite different patterns of use with different implications for glycaemic control and potentially different approaches to improving adherence.

Internationally, researchers have found that medication use among people with chronic conditions is often cyclical, with some people going through multiple discontinuations and reinitiations [11–15]. While this pattern has been observed among people taking metformin for T2DM in other countries [16, 17], the degree to which it occurs in New Zealand was unknown. To address this gap in knowledge, we undertook an analysis based on a national cohort of people with T2DM who initiated metformin monotherapy to explore the extent to which they continued with their prescribed therapy.

The primary aim of the study was to describe the patterns of discontinuation and reinitiation in new users of metformin monotherapy in New Zealand, overall and according to person- and healthcare-related factors. The secondary aim was to estimate the rate of discontinuation of metformin monotherapy, both overall and by person- and healthcare-related factors.

# 2. Materials and methods

This study was based on a cohort of people aged ≥ 18 years who initiated metformin monotherapy for T2DM in New Zealand between 1 January 2006 and 30 September 2014 and who had at least 455 days of follow-up. The methods for deriving this new user cohort, constructing the metformin possession record, and defining covariates are described in detail elsewhere [10]; we briefly summarise them below.

Where feasible, we have conformed to the EMERGE Guidelines for reporting medication adherence studies [18]. However, we do not use the taxonomy described by Vrijens et al. [19] and mandated by the EMERGE Guidelines because we feel that it is conceptually ambiguous when considering adherence as a dynamic process as we do here. In particular, the taxonomy does not address reinitiation of therapy after discontinuation. Instead, we have used the refill-gap method as a more appropriate framework for examining the dynamics of medication adherence [20].

## 2.1 Data sources

The source population was the Ministry of Health's Virtual Diabetes Register (VDR) [21, 22]. The VDR is an annually updated national register of people presumed to have diabetes mellitus who are identified using records of diabetes-related hospital admissions and outpatient visits, retinal screening, repeated HbA1c laboratory tests, and dispensings of antidiabetic medications. Women with gestational diabetes and those taking metformin for polycystic ovary syndrome are excluded. Past evaluations of the VDR have indicated it has good capture of people with diagnosed diabetes [23–25]. The VDR for any given year excludes patients who died or were not enrolled in a Primary Health Organisation during that year; however for this study we asked the Ministry to retain recently deceased and unenrolled patients in the VDR dataset.

For each individual included on the VDR between 1 January 2005 and 31 December 2014, the Ministry of Health provided demographic, health, and pharmaceutical dispensing data from its national collections.

## 2.2 Derivation of the study cohort

The steps we took to derive the study cohort are outlined in **S1 Fig**. In brief, we identified all individuals on the VDR who were dispensed metformin at least once between 1 January 2005 and 30 September 2014. The latter date was chosen to allow sufficient follow-up time between cohort entry (the date of the first metformin dispensing) and the end of the study period (31 December 2015). We opted to employ a new-user design so we could track medication use as a function of time from therapy initiation. We excluded people with a cohort entry date in 2005 as well as those with an initial metformin dose of more than 1,000mg per day (as new users of metformin are unlikely to start on such a high dose) to reduce the likelihood of including past users of metformin. Because our focus was on metformin monotherapy, we excluded people who had received a dispensing of any other antidiabetic medication before, or within 14 days after, cohort entry. As metformin is occasionally used in type 1 diabetes, we excluded people with hospital discharge or death records consistent with a type 1 diabetes diagnosis at any time before the end of the study period. We also excluded people for whom days' supply data were missing from all their metformin dispensing records, as this information was needed to calculate metformin possession dates. We chose to focus on adults with T2DM, so excluded those aged < 18 years at cohort entry. We excluded a very small number of people who did not appear to be normally resident in New Zealand at the time of the first metformin dispensing. Finally, we excluded people who died within 455 days of initiating metformin (because we required > 455 days of follow-up in order to calculate discontinuation in the first year) and people who permanently discontinued metformin < 100 days after initiation (as this group was likely to include people who had an intolerance to metformin).

Cohort members were followed until the earlier of their date of death or the end of the study period.

## 2.3 Construction of medication record

For each cohort member, we generated a metformin monotherapy possession record using community pharmacy dispensing data (dates, days' supply, tablet strength) from the national Pharmaceutical Collection (Pharms) [26]. Pharms contains details of community pharmacy dispensings of medications subsidised by the state (virtually all commonly used medications in New Zealand), providing a comprehensive information source on community medication use. Adjustments were made to correct for overlapping dispensings, simultaneous dispensings of different metformin tablet strengths, and missing values for the days supplied variable (see [10] for details).

## 2.4 Discontinuation and reinitiation measures

We defined discontinuation of metformin monotherapy as a gap of $\geq$ 90 days between the end of one dispensed supply and the earliest of (i) a subsequent dispensing of metformin monotherapy, (ii) dispensing of another antidiabetic pharmacological regimen (defined as a complete change in medication or the addition of another antidiabetic agent(s) to metformin), (iii) death, or (iv) the end of the study period.

We chose 90 days as this represents the maximum days of supply that can usually be dispensed at one time in New Zealand, making it unlikely that the gap was due to stockpiling. It is also sufficiently large to represent a clinically relevant lapse in treatment. For cohort members who discontinued, the first day on which they were no longer in possession of metformin monotherapy was recorded as the first day of the discontinuation period.

We defined reinitiation of metformin monotherapy as receiving a new supply after a discontinuation period of $\geq$ 90 days.

## 2.5 Covariates

We obtained information about person- and healthcare-related factors from the Ministry of Health's national collections. The baseline date for the analyses exploring the duration of the initial period of metformin monotherapy, and the rate of discontinuation of metformin monotherapy, was the date of cohort entry. Baseline dates for the analyses exploring the time to reinitiation after discontinuation and the duration of reinitiated metformin monotherapy were the date of discontinuation and the date of reinitiation, respectively.

We extracted, or derived, the following covariates for each cohort member: age at baseline; gender; self-identified ethnicity; socioeconomic deprivation at baseline; district health board (DHB) region of residence at baseline; Charlson Comorbidity Index (CCI, calculated based on hospitalisations in the 5 years before baseline); history of cardiovascular disease (CVD) at baseline; cancer registration and number of hospitalisations in the year before baseline; and, in the 6 months before baseline, evidence of diagnosed depression, number of non-diabetic medications, a glucose laboratory test (including HbA1c, fructosamine, glucose tolerance test and serum glucose), and a urinary albumin/creatinine ratio test. These covariates were chosen because they represented important population groups, were factors identified in the literature as potentially influencing metformin adherence, or were tests recommended in New Zealand as part of clinical monitoring for people with T2DM.

## 2.6 Statistical analyses

We used the refill-gap method as a framework for our analyses [20], along with time-to-event analyses as recommended by Vrijens et al. [19, 27]. We used competing risk methods to evaluate the duration of the first episode of metformin monotherapy, the time to first reinitiation of metformin monotherapy, and the duration of reinitiated use. Dispensing of another antidiabetic regimen and death were considered as competing risks, and follow-up was censored at the end of the study period. We calculated cumulative incidences (referred to as proportions) and 95% confidence intervals (95% CIs) for groups defined by person- and healthcare-related factors using the method of Choudhury [28] as implemented in the *stcompet* function in Stata version 15 [29]. We used competing risks regression to compare groups [30] and present sub-hazard ratios (referred to as hazard ratios [HRs]) and 95% CIs. We fitted both univariable and multivariable models, with multivariable models adjusting for the person- and healthcare-related factors described in section 2.5.

We calculated the rate of discontinuation during follow-up, overall and by person- and healthcare-related factors. The discontinuation rate is a function of whether a person

discontinues at all, how long they discontinue for, and how many times they discontinue and reinitiate. The rate is best viewed as an indicator of *adherence volatility* for a particular group, with a higher rate indicating more frequent transitions between discontinuation/reinitiation states. We censored follow-up at the earliest of a change to another antidiabetic regimen, death, or the end of the study period. We used Poisson regression to calculate the rates of discontinuation per 10 person-years, as well as crude and adjusted rate ratios (RRs) for the person- and healthcare-related factors, using the number of discontinuations during follow-up as the outcome and log follow-up time as the offset. Robust standard errors were calculated to allow for over-dispersion and were used to construct 95% CIs for the rates and RRs. We used R v4.0.1 [31] to perform these analyses.

Since the proportion of missing data for person- and healthcare-related factors was very low, we did not impute missing data.

## 2.7 Ethical approval

We obtained ethical approval from the University of Otago Human Research Ethics Committee (Health) (reference number HD17/027).

# 3. Results

## 3.1 Study cohort

The characteristics of the study cohort (n = 85,066) at entry are shown in **Table 1**. The median follow-up was 3.6 years [IQR: 2.0–5.8 years].

## 3.2 Discontinuation

During follow-up, 2,072 (2.4%) cohort members died while still being dispensed metformin monotherapy, 19,982 (23.5%) changed to another antidiabetic regimen, and 24,165 (28.4%) continued to take metformin monotherapy through to the end of the study period. This left 38,847 (45.7%) who discontinued metformin monotherapy at least once.

**Fig 1** shows the cumulative incidence of the first discontinuation of metformin monotherapy, of changing to another antidiabetic regimen, and of death. At year 1 after cohort entry, 28.2% of cohort members had discontinued metformin monotherapy at least once, 9.0% had changed to another antidiabetic regimen, and 62.8% had continued to take metformin monotherapy; by design, none had died. By the end of year 2, 36.8% had discontinued metformin monotherapy at least once, 14.1% had changed to another antidiabetic regimen, 0.6% had died without discontinuing metformin monotherapy, and 48.5% remained alive and had continued metformin monotherapy; the corresponding figures at 5 years were 46.3%, 23.7%, 2.3%, and 27.7%.

The cumulative proportions of cohort members who discontinued metformin monotherapy at least once differed according to person- and healthcare-related factors (**Table 1**). Some groups already disproportionately affected by T2DM (Māori, Pacific, and Indian ethnic groups and those living in socioeconomically deprived areas) were more likely to discontinue. There was a marked inverse relationship between increasing age and discontinuation, and cohort members with poorer health status at cohort entry, such as a higher CCI, a history of CVD, a cancer registration, depression, and higher numbers of hospitalisations and non-diabetic medications were less likely to discontinue. Similarly, cohort members with a recent glucose or a urinary albumin/creatinine ratio test were less likely to discontinue than those who were not tested. The proportions who discontinued varied according to DHB, while there was little difference by gender.

**Table 1. Comparison of cumulative proportions who discontinued metformin monotherapy at least once by person- and healthcare-related factors.**

| Person- or healthcare-related factor | Study cohort N (%) | Cumulative proportion (95% CI) who discontinued | | | Hazard ratio | |
|---|---|---|---|---|---|---|
| | | End of year 1 | End of year 2 | End of year 5 | Unadjusted (95% CI) | Adjusted* (95% CI) |
| **Age at cohort entry (years)** | | | | | | |
| <25 | 752 (0.9) | 61.7 (58.1–65.1) | 70.4 (66.9–73.5) | 74.6 (71.3–77.7) | 1.64 (1.49–1.80) | 1.46 (1.32–1.61) |
| 25–34 | 3,613 (4.2) | 54.1 (52.5–55.7) | 63.5 (61.9–65.0) | 69.5 (67.9–71.0) | 1.34 (1.28–1.41) | 1.24 (1.18–1.30) |
| 35–44 | 10,965 (12.9) | 42.2 (41.3–43.2) | 52.0 (51.1–53.0) | 60.7 (59.7–61.6) | Reference | Reference |
| 45–54 | 21,434 (25.2) | 32.6 (32.0–33.3) | 41.9 (41.3–42.6) | 51.2 (50.5–52.0) | 0.76 (0.73–0.78) | 0.82 (0.80–0.85) |
| 55–64 | 23,292 (27.4) | 23.9 (23.3–24.4) | 32.2 (31.6–32.8) | 41.6 (40.9–42.2) | 0.57 (0.55–0.58) | 0.69 (0.67–0.71) |
| 65–74 | 16,670 (19.6) | 18.2 (17.6–18.8) | 25.5 (24.9–26.2) | 35.7 (34.9–36.4) | 0.46 (0.45–0.48) | 0.63 (0.61–0.66) |
| ≥75 | 8,340 (9.8) | 16.0 (15.2–16.8) | 24.0 (23.1–25.0) | 37.0 (35.9–38.1) | 0.47 (0.46–0.49) | 0.73 (0.70–0.77) |
| **Gender** | | | | | | |
| Female | 40,140 (47.2) | 28.7 (28.3–29.1) | 37.6 (37.1–38.1) | 47.5 (47.0–48.0) | Reference | Reference |
| Male | 44,926 (52.8) | 27.7 (27.3–28.1) | 36.0 (35.6–36.5) | 45.3 (44.8–45.8) | 0.94 (0.92–0.95) | 0.93 (0.92–0.95) |
| **Ethnicity (prioritised)†‡** | | | | | | |
| Māori | 13,596 (16.0) | 37.1 (36.3–37.9) | 46.5 (45.7–47.4) | 55.6 (54.7–56.4) | 1.71 (1.66–1.76) | 1.48 (1.44–1.53) |
| Pacific | 11,135 (13.1) | 44.3 (43.3–45.2) | 53.2 (52.2–54.1) | 60.8 (59.8–61.7) | 2.03 (1.97–2.09) | 1.58 (1.52–1.63) |
| European | 44,578 (52.4) | 20.5 (20.1–20.8) | 28.4 (28.0–28.8) | 38.6 (38.1–39.0) | Reference | Reference |
| Asian (Non-Indian) | 5,969 (7.0) | 29.7 (28.5–30.8) | 39.1 (37.8–40.3) | 49.3 (47.9–50.6) | 1.39 (1.33–1.44) | 1.13 (1.09–1.18) |
| Indian | 5,536 (6.5) | 34.8 (33.6–36.1) | 44.6 (43.2–45.9) | 53.6 (52.2–55.0) | 1.61 (1.55–1.67) | 1.25 (1.20–1.31) |
| Other | 1,187 (1.4) | 30.2 (27.6–32.8) | 41.2 (38.4–44.0) | 53.2 (50.1–56.2) | 1.52 (1.41–1.65) | 1.25 (1.15–1.35) |
| **Socioeconomic deprivation (NZDep13) at cohort entry‡** | | | | | | |
| Quintile 1 | 10,308 (12.1) | 24.5 (23.7–25.4) | 33.1 (32.2–34.0) | 43.3 (42.2–44.3) | Reference | Reference |
| Quintile 2 | 12,002 (14.1) | 24.6 (23.9–25.4) | 33.6 (32.8–34.5) | 43.8 (42.9–44.8) | 1.03 (0.99–1.08) | 0.97 (0.94–1.02) |
| Quintile 3 | 15,054 (17.7) | 25.0 (24.3–25.7) | 33.2 (32.5–34.0) | 43.4 (42.6–44.3) | 1.04 (1.00–1.08) | 0.98 (0.94–1.02) |
| Quintile 4 | 19,427 (22.8) | 26.6 (26.0–27.3) | 35.2 (34.5–35.9) | 44.9 (44.1–45.6) | 1.10 (1.06–1.14) | 1.00 (0.96–1.04) |
| Quintile 5 | 28,271 (33.2) | 33.8 (33.2–34.3) | 42.4 (41.8–43.0) | 51.1 (50.5–51.7) | 1.39 (1.34–1.44) | 1.03 (0.99–1.07) |
| **District Health Board at cohort entry‡** | | | | | | |
| Auckland | 9,453 (11.1) | 34.4 (33.5–35.4) | 44.0 (43.0–45.0) | 53.8 (52.8–54.9) | Reference | Reference |
| Bay of Plenty | 3,665 (4.3) | 24.9 (23.6–26.4) | 32.6 (31.1–34.1) | 42.3 (40.7–44.0) | 0.72 (0.68–0.76) | 0.80 (0.76–0.85) |
| Canterbury | 7,634 (9.0) | 24.2 (23.2–25.2) | 32.7 (31.6–33.7) | 43.3 (42.1–44.5) | 0.73 (0.7–0.76) | 0.87 (0.83–0.91) |
| Capital and Coast | 4,505 (5.3) | 25.7 (24.4–27.0) | 34.2 (32.9–35.6) | 42.0 (40.5–43.5) | 0.72 (0.68–0.76) | 0.75 (0.71–0.80) |
| Counties Manukau | 14,700 (17.3) | 35.5 (34.7–36.3) | 44.4 (43.6–45.2) | 53.3 (52.4–54.1) | 1.00 (0.97–1.04) | 0.95 (0.92–0.99) |
| Hawkes Bay | 3,403 (4.0) | 25.7 (24.2–27.1) | 34.2 (32.6–35.8) | 45.0 (43.3–46.8) | 0.79 (0.74–0.83) | 0.85 (0.80–0.91) |
| Hutt | 2,811 (3.3) | 25.1 (23.5–26.7) | 33.8 (32.1–35.6) | 41.6 (39.7–43.5) | 0.71 (0.66–0.75) | 0.75 (0.70–0.80) |
| Lakes | 1,880 (2.2) | 28.2 (26.2–30.2) | 35.7 (33.6–37.9) | 44.9 (42.5–47.3) | 0.78 (0.73–0.84) | 0.84 (0.78–0.90) |
| MidCentral | 3,466 (4.1) | 23.9 (22.5–25.3) | 31.8 (30.2–33.3) | 41.5 (39.7–43.2) | 0.69 (0.65–0.73) | 0.81 (0.76–0.87) |
| Nelson Marlborough | 1,864 (2.2) | 22.5 (20.7–24.5) | 31.2 (29.1–33.3) | 42.0 (39.6–44.4) | 0.69 (0.64–0.74) | 0.82 (0.76–0.89) |
| Northland | 3,440 (4.0) | 27.5 (26.1–29.0) | 36.6 (35.0–38.3) | 46.7 (44.9–48.4) | 0.81 (0.77–0.86) | 0.88 (0.83–0.93) |
| South Canterbury | 1,066 (1.3) | 21.3 (18.9–23.8) | 29.1 (26.4–31.9) | 38.0 (34.9–41.0) | 0.62 (0.56–0.68) | 0.80 (0.72–0.89) |
| Southern | 4,939 (5.8) | 20.1 (19.0–21.2) | 27.9 (26.6–29.1) | 38.4 (37.0–39.8) | 0.62 (0.59–0.65) | 0.80 (0.76–0.85) |
| Tairawhiti | 999 (1.2) | 31.5 (28.7–34.4) | 39.6 (36.5–42.6) | 49.4 (46.1–52.6) | 0.90 (0.82–0.99) | 0.90 (0.82–0.99) |
| Taranaki | 2,049 (2.4) | 20.6 (18.8–22.3) | 28.1 (26.2–30.1) | 39.2 (37.0–41.5) | 0.65 (0.60–0.70) | 0.82 (0.76–0.88) |
| Waikato | 6,355 (7.5) | 27.0 (25.9–28.1) | 35.6 (34.4–36.8) | 43.8 (42.5–45.1) | 0.75 (0.72–0.79) | 0.87 (0.83–0.92) |
| Wairarapa | 857 (1.0) | 22.1 (19.3–24.9) | 31.5 (28.4–34.6) | 42.4 (38.9–45.8) | 0.71 (0.64–0.78) | 0.86 (0.77–0.95) |
| Waitemata | 10,103 (11.9) | 29.1 (28.2–30.0) | 38.0 (37.0–39.0) | 47.7 (46.7–48.7) | 0.84 (0.81–0.87) | 0.94 (0.90–0.98) |
| West Coast | 472 (0.6) | 21.6 (18.0–25.4) | 28.9 (24.8–33.0) | 41.3 (36.5–46.0) | 0.66 (0.58–0.76) | 0.82 (0.71–0.95) |
| Whanganui | 1,396 (1.6) | 26.9 (24.6–29.2) | 34.0 (31.5–36.5) | 43.7 (40.9–46.4) | 0.74 (0.68–0.81) | 0.85 (0.78–0.93) |
| **Charlson comorbidity index at cohort entry** | | | | | | |
| 0 | 73,315 (86.2) | 29.5 (29.1–29.8) | 38.2 (37.8–38.5) | 47.6 (47.2–47.9) | Reference | Reference |
| 1 | 8,125 (9.6) | 21.3 (20.4–22.2) | 29.0 (28.1–30.0) | 39.4 (38.3–40.6) | 0.77 (0.74–0.79) | 0.97 (0.93–1.01) |
| 2 | 2,662 (3.1) | 17.8 (16.4–19.3) | 25.9 (24.3–27.6) | 37.0 (35.1–39.0) | 0.69 (0.65–0.73) | 0.94 (0.88–1.00) |
| ≥3 | 964 (1.1) | 17.0 (14.7–19.5) | 27.1 (24.4–30.0) | 37.4 (34.2–40.6) | 0.71 (0.64–0.78) | 1.02 (0.92–1.14) |

*(Continued)*

**Table 1.** (Continued)

| Person- or healthcare-related factor | Study cohort N (%) | Cumulative proportion (95% CI) who discontinued | | | Hazard ratio | |
|---|---|---|---|---|---|---|
| | | End of year 1 | End of year 2 | End of year 5 | Unadjusted (95% CI) | Adjusted* (95% CI) |
| **History of cardiovascular disease at cohort entry** | | | | | | |
| No | 64,558 (75.9) | 31.2 (30.9–31.6) | 40.1 (39.7–40.5) | 49.4 (49.0–49.8) | Reference | Reference |
| Yes | 20,508 (24.1) | 18.6 (18.1–19.2) | 26.2 (25.6–26.8) | 36.6 (35.9–37.3) | 0.66 (0.64–0.67) | 0.86 (0.84–0.89) |
| **Cancer registration in the year before cohort entry** | | | | | | |
| No | 84,561 (99.4) | 28.2 (27.9–28.5) | 36.8 (36.5–37.2) | 46.4 (46.0–46.8) | Reference | Reference |
| Yes | 505 (0.6) | 18.6 (15.4–22.1) | 26.1 (22.3–30.0) | 35.5 (31.1–39.9) | 0.72 (0.62–0.83) | 0.87 (0.75–1.01) |
| **Number of hospitalisations in the year before cohort entry** | | | | | | |
| 0 | 68,662 (80.7) | 28.6 (28.2–28.9) | 37.2 (36.8–37.5) | 46.6 (46.3–47.0) | Reference | Reference |
| 1 | 10,797 (12.7) | 27.5 (26.7–28.4) | 36.0 (35.1–36.9) | 45.9 (44.9–46.9) | 0.98 (0.95–1.01) | 1.08 (1.05–1.12) |
| 2–4 | 5,054 (5.9) | 24.9 (23.7–26.1) | 33.3 (32.0–34.6) | 43.5 (42.0–44.9) | 0.73 (0.51–1.05) | 1.01 (0.70–1.47) |
| 5–9 | 480 (0.6) | 26.5 (22.6–30.5) | 34.5 (30.3–38.8) | 43.1 (38.3–47.8) | 0.90 (0.86–0.94) | 1.11 (1.06–1.17) |
| ≥10 | 73 (0.1) | 19.2 (11.1–28.9) | 28.9 (19.0–39.5) | 37.0 (25.3–48.7) | 0.89 (0.78–1.03) | 1.29 (1.11–1.49) |
| **Depression in the 6 months before cohort entry** | | | | | | |
| No | 77,807 (91.5) | 28.7 (28.4–29.0) | 37.2 (36.9–37.6) | 46.6 (46.3–47.0) | Reference | Reference |
| Yes | 7,259 (8.5) | 22.3 (21.4–23.3) | 32.0 (30.9–33.1) | 43.0 (41.8–44.3) | 0.86 (0.83–0.90) | 1.03 (0.99–1.07) |
| **Number of non-diabetic medications used in the 6 months before cohort entry** | | | | | | |
| 0–1 | 9,207 (10.8) | 42.9 (41.8–43.9) | 52.0 (51.0–53.0) | 59.3 (58.3–60.4) | Reference | Reference |
| 2–3 | 17,443 (20.5) | 35.5 (34.8–36.2) | 44.8 (44.0–45.5) | 53.2 (52.5–54.0) | 0.82 (0.8–0.85) | 0.87 (0.84–0.90) |
| 4–5 | 18,443 (21.7) | 28.8 (28.1–29.4) | 37.5 (36.8–38.2) | 47.5 (46.7–48.2) | 0.68 (0.66–0.71) | 0.77 (0.75–0.80) |
| 6–7 | 14,482 (17.0) | 24.8 (24.1–25.5) | 33.0 (32.3–33.8) | 42.7 (41.9–43.6) | 0.59 (0.57–0.61) | 0.70 (0.67–0.73) |
| 8–9 | 10,074 (11.8) | 21.1 (20.3–21.9) | 29.1 (28.2–30.0) | 39.6 (38.6–40.6) | 0.53 (0.51–0.55) | 0.64 (0.61–0.67) |
| 10–19 | 14,466 (17.0) | 18.5 (17.9–19.2) | 26.6 (25.9–27.3) | 37.4 (36.6–38.3) | 0.49 (0.47–0.51) | 0.61 (0.58–0.64) |
| ≥20 | 951 (1.1) | 14.5 (12.4–16.8) | 23.0 (20.4–25.8) | 33.4 (30.1–36.7) | 0.41 (0.37–0.46) | 0.51 (0.45–0.58) |
| **Glucose test in the 6 months before cohort entry§** | | | | | | |
| No | 11,127 (13.1) | 37.2 (36.3–38.1) | 46.5 (45.6–47.4) | 54.9 (53.9–55.8) | Reference | Reference |
| Yes | 73,939 (86.9) | 26.8 (26.5–27.1) | 35.3 (35.0–35.7) | 45.0 (44.7–45.4) | 0.73 (0.71–0.75) | 0.86 (0.83–0.89) |
| **Urinary albumin/creatinine ratio test in the 6 months before cohort entry** | | | | | | |
| No | 49,632 (58.3) | 29.4 (29.0–29.8) | 38.4 (37.9–38.8) | 48.4 (47.9–48.9) | Reference | Reference |
| Yes | 35,434 (41.7) | 26.5 (26.1–27.0) | 34.5 (34.0–35.0) | 43.4 (42.9–44.0) | 0.87 (0.85–0.89) | 0.89 (0.87–0.91) |

* Adjusted for all other covariates in the table.

† In the New Zealand healthcare system, people can record up to six ethnic groups. For statistical purposes, each individual can be allocated–a single ethnic group using a prioritisation algorithm [32]. The MELAA group (Middle Eastern, Latin American, African) was included in Other.

‡ Of the 85,066 people included in this analysis, ethnicity was unknown for 3,065, NZDep13 was unknown for 4, and District Health Board was unknown for 9.

§ Record of a laboratory test in the 'blood glucose' category (includes HbA1c, fructosamine, glucose tolerance, and serum glucose tests).

After adjustment for person- and healthcare-related factors in the multivariate regression model (**Table 1**), the differences in discontinuation by age, ethnicity, history of CVD, number of non-diabetic medications, DHB, and having a glucose test remained. However, there was no longer an association between discontinuation and deprivation, CCI, cancer registration, depression, or having a urinary albumin/creatinine ratio test. Increasing numbers of hospital admissions were associated with a higher likelihood of discontinuation in the adjusted model, but this could be reflecting the strong positive association between non-diabetic medications and the number of hospitalisations and the resulting collinearity.

### 3.3 Reinitiation

Of the 38,847 cohort members who discontinued metformin monotherapy, 14 were excluded from the reinitiation analysis as they were recorded as living overseas at the time. Of the

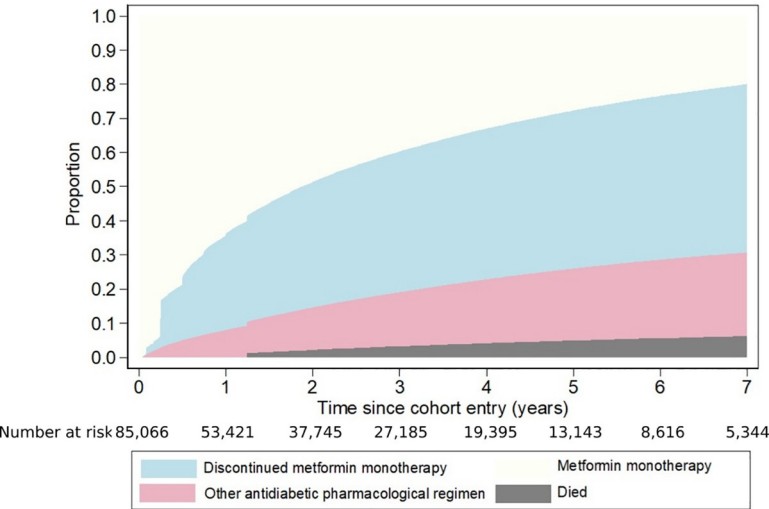

**Fig 1. Cumulative proportions who discontinued metformin monotherapy at least once, who changed to another antidiabetic regimen, and who died.**

remaining 38,833, 887 (2.3%) subsequently died without restarting any antidiabetic medication, 2,008 (5.2%) started another antidiabetic regimen, and 7,310 (18.8%) were alive but had not restarted any antidiabetic medication by the end of follow-up. This left 28,628 (73.7%) who reinitiated metformin monotherapy during follow-up.

Fig 2 shows the cumulative incidence of metformin monotherapy reinitiation, of starting another antidiabetic regimen, and of death. One year after the first discontinuation, 22.8% had reinitiated metformin monotherapy, 1.3% had started another antidiabetic regimen, and three quarters (75.9%) had not resumed pharmacological therapy. By the end of year 2, the proportions of cohort members who had reinitiated metformin monotherapy and started another antidiabetic regimen had increased to 49.0% and 3.0%, respectively, while 0.3% had died without restarting pharmacological therapy and a further 47.7% were still alive but had not

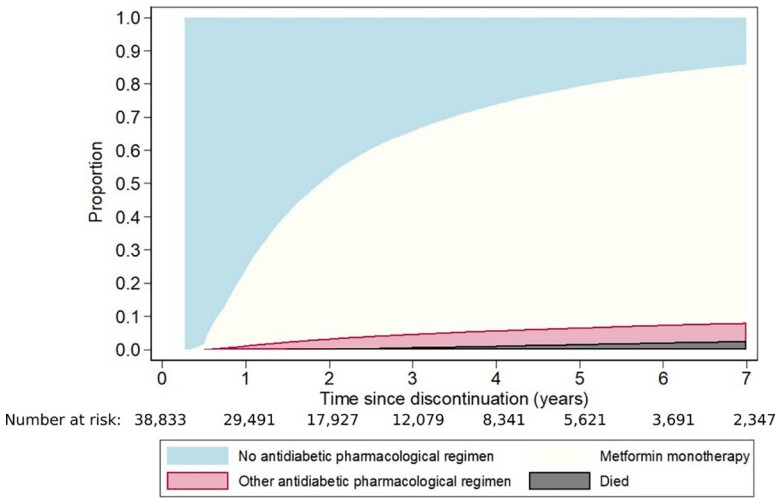

**Fig 2. Cumulative proportions who reinitiated metformin monotherapy after the first discontinuation, who changed to another antidiabetic regimen, and who died.**

reinitiated any antidiabetic medication. The corresponding figures at 5 years were 72.6%, 5.0%, 1.7%, and 20.7%.

The cumulative proportions reinitiating metformin monotherapy after the first discontinuation varied by person- and healthcare-related factors (**Table 2**). The groups disproportionately affected by T2DM who were more likely to discontinue (Māori, Pacific, and Indian cohort members and those living in socioeconomically deprived areas) were also more likely to reinitiate use. Similarly, those DHBs where people were most likely to discontinue were also those where people were most likely to reinitiate use. In contrast, people in the older age groups were less likely to discontinue metformin monotherapy but, when they did, were also less likely to reinitiate it. We also found this for cohort members with poorer health, who were less likely to discontinue; in general, those with poorer health who had discontinued were less likely to reinitiate metformin monotherapy than those with better health. For instance, people with a higher CCI, a history of CVD, a cancer registration, depression, and higher numbers of hospitalisations and non-diabetic medications were less likely to reinitiate. Finally, those with a recent glucose or urinary albumin/creatinine ratio test were more likely to reinitiate.

The differences in reinitiation by age and ethnicity remained after adjustment for person- and healthcare-related factors in the multivariate regression model (**Table 2**), although they were attenuated, as were the differences by DHB. There was little difference in reinitiation by socioeconomic deprivation. The general finding that people with poorer health at discontinuation were less likely to reinitiate persisted after adjustment, as did the association between receiving glucose or albumin/creatinine ratio tests before discontinuation and an increased likelihood of reinitiation.

### 3.4 Second discontinuation

Of those who reinitiated metformin monotherapy after the first discontinuation, nearly half (47.6%) had discontinued at least once in the first year following reinitiation (**S2 Fig**). The cumulative proportions who discontinued varied by person- and healthcare-related factors, with patterns similar to the first discontinuation (**S1 Table**).

### 3.5 Rates of discontinuation

The overall rate of discontinuations per 10 person-years was 1.98 (95% CI 1.96 to 1.99). The rates, and crude and adjusted RRs, by person- and healthcare-related factors are provided in **Table 3**. There were strong inverse relationships between increasing age and increasing numbers of non-diabetic medications dispensed in the 6 months before cohort entry and the rates of discontinuation. There were also differences between ethnic groups; compared with Europeans, Māori, Pacific, and Indian groups had higher rates of discontinuation. Increasing rates were also observed with increasing deprivation, although this relationship largely disappeared after adjustment for other person- and healthcare-related factors.

## 4. Discussion

Our findings from this national cohort study highlight the dynamic nature of metformin monotherapy use in New Zealand and show that some people have substantial periods of non-use. For instance, after 2 years of follow-up, just over a third of cohort members had discontinued metformin monotherapy at least once without starting another antidiabetic medication, dying or reaching the end of follow-up. After 5 years, almost half of those still in follow-up had discontinued metformin monotherapy at least once. Of those who discontinued, three quarters did not restart any antidiabetic medication in the year following their first discontinuation. However, after 2 years 49% had reinitiated metformin monotherapy and 3% had started

**Table 2. Comparison of cumulative proportions who reinitiated metformin monotherapy after first discontinuation by person- and healthcare-related factors.**

| Person- or healthcare-related factor | | Cumulative proportion (95% CI) who reinitiated | | | Hazard ratio | |
|---|---|---|---|---|---|---|
| | | End of year 1 | End of year 2 | End of year 5 | Unadjusted (95% CI) | Adjusted* (95% CI) |
| **Age at discontinuation (years)** | | | | | | |
| | <25 | 25.8 (22.0–29.7) | 52.5 (47.9–56.9) | 77.5 (73.1–81.2) | 0.82 (0.74–0.92) | 0.84 (0.75–0.95) |
| | 25–34 | 28.1 (26.3–30.0) | 60 (57.9–62.0) | 81.8 (80.0–83.4) | 0.97 (0.91–1.02) | 0.96 (0.90–1.01) |
| | 35–44 | 29.4 (28.2–30.5) | 61 (59.8–62.3) | 82.4 (81.3–83.4) | Reference | Reference |
| | 45–54 | 27.7 (26.9–28.6) | 56.6 (55.6–57.5) | 80.6 (79.7–81.4) | 0.92 (0.89–0.96) | 0.97 (0.94–1.01) |
| | 55–64 | 21.6 (20.8–22.5) | 48.7 (47.7–49.7) | 75.1 (74.1–76.0) | 0.77 (0.74–0.79) | 0.87 (0.83–0.90) |
| | 65–74 | 16.2 (15.3–17.1) | 37.4 (36.2–38.6) | 63.0 (61.7–64.2) | 0.55 (0.53–0.58) | 0.68 (0.65–0.72) |
| | ≥75 | 10.2 (9.3–11.2) | 23.5 (22.2–24.9) | 41.7 (40.1–43.3) | 0.31 (0.29–0.33) | 0.46 (0.43–0.49) |
| **Gender** | | | | | | |
| | Female | 21.9 (21.3–22.5) | 47.0 (46.3–47.7) | 71.0 (70.4–71.7) | Reference | Reference |
| | Male | 23.7 (23.1–24.3) | 50.8 (50.1–51.5) | 74.1 (73.5–74.8) | 1.10 (1.07–1.13) | 1.07 (1.05–1.10) |
| **Ethnicity (prioritised)[†] [‡]** | | | | | | |
| | Māori | 27.4 (26.4–28.4) | 55.9 (54.8–57.0) | 79.9 (78.9–80.8) | 1.57 (1.53–1.62) | 1.31 (1.27–1.36) |
| | Pacific | 31.2 (30.1–32.3) | 62.9 (61.7–64.0) | 84.3 (83.4–85.2) | 1.85 (1.79–1.91) | 1.40 (1.35–1.46) |
| | European | 17.5 (16.9–18.1) | 39.7 (38.9–40.4) | 63.7 (63.0–64.5) | Reference | Reference |
| | Asian (Non-Indian) | 21.8 (20.3–23.3) | 50.0 (48.1–51.8) | 73.8 (72.1–75.5) | 1.32 (1.26–1.39) | 1.07 (1.02–1.12) |
| | Indian | 25.4 (23.8–27.0) | 54.7 (52.8–56.5) | 79.2 (77.6–80.7) | 1.53 (1.47–1.60) | 1.18 (1.13–1.24) |
| | Other | 20.0 (17.0–23.3) | 44.8 (40.9–48.7) | 72.3 (68.3–76.0) | 1.22 (1.11–1.34) | 0.98 (0.89–1.08) |
| **Socioeconomic deprivation (NZDep13) at discontinuation[‡]** | | | | | | |
| | Quintile 1 | 20.8 (19.6–22.0) | 45.0 (43.5–46.5) | 69.8 (68.4–71.2) | Reference | Reference |
| | Quintile 2 | 20.5 (19.4–21.5) | 45.1 (43.7–46.4) | 69.8 (68.4–71.1) | 1.01 (0.96–1.06) | 1.00 (0.95–1.05) |
| | Quintile 3 | 20.5 (19.5–21.5) | 45.3 (44.1–46.5) | 70.1 (68.9–71.2) | 1.01 (0.96–1.05) | 1.00 (0.95–1.05) |
| | Quintile 4 | 21.3 (20.4–22.2) | 47.5 (46.4–48.5) | 70.5 (69.5–71.5) | 1.05 (1.01–1.10) | 1.01 (0.97–1.06) |
| | Quintile 5 | 26.4 (25.6–27.1) | 54.4 (53.6–55.2) | 77.2 (76.5–77.9) | 1.29 (1.24–1.34) | 1.05 (1.01–1.10) |
| **District Health Board at discontinuation[‡]** | | | | | | |
| | Auckland | 24.6 (23.4–25.8) | 53.2 (51.8–54.6) | 76.9 (75.6–78.1) | Reference | Reference |
| | Bay of Plenty | 21.7 (19.7–23.8) | 45.6 (43.1–48.1) | 69.8 (67.3–72.1) | 0.82 (0.77–0.88) | 0.95 (0.89–1.02) |
| | Canterbury | 20.0 (18.7–21.4) | 45.2 (43.5–46.9) | 69.4 (67.7–71.0) | 0.82 (0.78–0.86) | 1.01 (0.96–1.07) |
| | Capital and Coast | 22.1 (20.2–24.0) | 48.2 (45.9–50.5) | 71.6 (69.4–73.8) | 0.87 (0.81–0.92) | 0.89 (0.83–0.95) |
| | Counties Manukau | 27.6 (26.6–28.6) | 56.0 (54.8–57.1) | 79.3 (78.4–80.3) | 1.08 (1.04–1.13) | 1.00 (0.95–1.04) |
| | Hawkes Bay | 20.5 (18.5–22.6) | 45.0 (42.5–47.5) | 67.9 (65.4–70.3) | 0.79 (0.74–0.85) | 0.87 (0.81–0.93) |
| | Hutt | 24.1 (21.6–26.6) | 49.6 (46.7–52.5) | 73.3 (70.5–75.9) | 0.92 (0.85–0.99) | 0.96 (0.89–1.04) |
| | Lakes | 25.9 (22.9–28.9) | 49.6 (46.1–52.9) | 73.1 (69.7–76.1) | 0.90 (0.82–0.98) | 0.93 (0.85–1.02) |
| | MidCentral | 19.3 (17.2–21.4) | 43.4 (40.7–46.0) | 67.2 (64.4–69.8) | 0.75 (0.70–0.81) | 0.87 (0.81–0.94) |
| | Nelson Marlborough | 17.2 (14.7–20.0) | 42.0 (38.5–45.5) | 64.7 (61.0–68.2) | 0.71 (0.64–0.77) | 0.91 (0.82–1.00) |
| | Northland | 19.7 (17.7–21.7) | 45.1 (42.6–47.6) | 69.3 (66.8–71.6) | 0.80 (0.75–0.86) | 0.84 (0.78–0.90) |
| | South Canterbury | 18.8 (15.2–22.8) | 39.7 (34.8–44.5) | 62.6 (57.4–67.4) | 0.68 (0.59–0.77) | 0.96 (0.84–1.10) |
| | Southern | 17.7 (16.0–19.5) | 39.7 (37.5–42.0) | 65.6 (63.3–67.9) | 0.72 (0.68–0.77) | 0.95 (0.89–1.02) |
| | Tairawhiti | 23.2 (19.6–27.0) | 47.9 (43.4–52.3) | 73.2 (68.8–77.1) | 0.88 (0.79–0.98) | 0.89 (0.79–0.99) |
| | Taranaki | 20.0 (17.3–22.8) | 40.8 (37.4–44.2) | 64.1 (60.6–67.4) | 0.71 (0.65–0.78) | 0.86 (0.79–0.95) |
| | Waikato | 22.6 (21.1–24.2) | 50.1 (48.2–52) | 73.7 (71.8–75.4) | 0.91 (0.86–0.96) | 0.98 (0.93–1.04) |
| | Wairarapa | 19.6 (15.6–23.8) | 40.9 (35.7–46) | 66.2 (60.7–71.1) | 0.74 (0.65–0.84) | 0.88 (0.77–1.01) |
| | Waitemata | 23.1 (21.9–24.3) | 51.4 (49.9–52.8) | 75.5 (74.2–76.8) | 0.95 (0.91–1.00) | 1.03 (0.98–1.08) |
| | West Coast | 16.0 (10.9–21.9) | 33.8 (26.8–40.9) | 58.7 (50.8–65.8) | 0.59 (0.49–0.72) | 0.82 (0.68–0.99) |
| | Whanganui | 22.6 (19.3–26.0) | 45.2 (41.1–49.2) | 72.0 (68.0–75.6) | 0.86 (0.78–0.94) | 0.97 (0.88–1.07) |
| **Charlson Comorbidity Index at discontinuation** | | | | | | |

(Continued)

**Table 2.** (Continued)

| Person- or healthcare-related factor | | Cumulative proportion (95% CI) who reinitiated | | | Hazard ratio | |
|---|---|---|---|---|---|---|
| | | End of year 1 | End of year 2 | End of year 5 | Unadjusted (95% CI) | Adjusted* (95% CI) |
| | 0 | 23.8 (23.4–24.3) | | 75.3 (74.8–75.8) | Reference | Reference |
| | 1 | 18.9 (17.6–20.2) | 51.1 (50.5–51.6) | 63.5 (61.8–65.1) | 0.73 (0.70–0.76) | 1.03 (0.98–1.08) |
| | 2 | 13.4 (11.6–15.4) | 40.7 (39.1–42.3) | 48.3 (45.3–51.1) | 0.48 (0.44–0.52) | 0.84 (0.77–0.92) |
| | ≥3 | 8.9 (6.9–11.2) | 30.1 (27.5–32.6) | 35.2 (31.5–38.9) | 0.31 (0.27–0.35) | 0.67 (0.59–0.77) |
| **History of cardiovascular disease at discontinuation** | | | | | | |
| | No | 25 (24.5–25.5) | 53.4 (52.9–54.0) | 77.1 (76.6–77.6) | Reference | Reference |
| | Yes | 15.7 (15–16.5) | 34.7 (33.7–35.6) | 58.6 (57.5–59.6) | 0.60 (0.58–0.62) | 0.85 (0.82–0.88) |
| **Cancer registration in the year before discontinuation** | | | | | | |
| | No | 22.9 (22.5–23.4) | 49.2 (48.7–49.7) | 72.9 (72.4–73.4) | Reference | Reference |
| | Yes | 10.1 (7.4–13.2) | 27.4 (23.2–31.7) | 47.6 (42.6–52.4) | 0.50 (0.44–0.57) | 0.89 (0.77–1.02) |
| **Number of hospitalisations in the year before discontinuation** | | | | | | |
| | 0 | 24.1 (23.6–24.6) | 51.2 (50.6–51.8) | 75.3 (74.8–75.8) | Reference | Reference |
| | 1 | 20.8 (19.8–21.9) | 46.7 (45.4–48.1) | 69.6 (68.2–70.8) | 0.85 (0.82–0.88) | 0.99 (0.96–1.03) |
| | 2–4 | 15.6 (14.3–16.9) | 34.9 (33.2–36.6) | 56.5 (54.7–58.4) | 0.59 (0.57–0.62) | 0.87 (0.82–0.92) |
| | 5–9 | 10.6 (8–13.7) | 25.5 (21.5–29.6) | 41.9 (37.2–46.6) | 0.40 (0.35–0.46) | 0.80 (0.69–0.92) |
| | ≥10 | 6.4 (2.1–14.2) | 17.5 (9.3–27.8) | 36.8 (24.7–49.0) | 0.31 (0.21–0.46) | 0.60 (0.40–0.92) |
| **Depression in the 6 months before discontinuation** | | | | | | |
| | No | 23.5 (23.1–24.0) | 50.2 (49.6–50.7) | 73.7 (73.2–74.2) | Reference | Reference |
| | Yes | 15.3 (14.1–16.5) | 36.4 (34.8–38.1) | 61.5 (59.7–63.2) | 0.70 (0.67–0.73) | 0.87 (0.83–0.91) |
| **Number of non-diabetic medications in the 6 months before discontinuation** | | | | | | |
| | 0–1 | 25.9 (24.7–27.1) | 54 (52.6–55.4) | 76.9 (75.6–78.1) | Reference | Reference |
| | 2–3 | 26.3 (25.3–27.2) | 56.7 (55.7–57.8) | 80.3 (79.4–81.2) | 1.08 (1.04–1.13) | 1.07 (1.03–1.12) |
| | 4–5 | 25.1 (24.2–26.0) | 53.0 (52.0–54.1) | 77.4 (76.5–78.4) | 0.99 (0.95–1.03) | 1.03 (0.98–1.07) |
| | 6–7 | 22.9 (21.9–23.9) | 48.8 (47.6–50.1) | 74.0 (72.8–75.1) | 0.90 (0.86–0.94) | 1.01 (0.97–1.06) |
| | 8–9 | 20.7 (19.4–22) | 44.8 (43.2–46.3) | 68.5 (67.0–70.0) | 0.78 (0.75–0.82) | 0.97 (0.92–1.02) |
| | 10–19 | 14.9 (14–15.8) | 33.8 (32.6–35.0) | 56.4 (55.0–57.7) | 0.56 (0.53–0.58) | 0.82 (0.78–0.86) |
| | ≥20 | 7.7 (5.6–10.1) | 18.5 (15.4–21.9) | 36.9 (32.7–41.1) | 0.31 (0.27–0.35) | 0.61 (0.53–0.70) |
| **Glucose test in the 6 months before discontinuation§** | | | | | | |
| | No | 15.9 (15.3–16.6) | 43.2 (42.3–44.1) | 69.4 (68.5–70.3) | Reference | Reference |
| | Yes | 25.7 (25.1–26.2) | 51.4 (50.8–52.0) | 74.0 (73.4–74.5) | 1.20 (1.17–1.23) | 1.23 (1.19–1.27) |
| **Urinary albumin/creatinine ratio test in the 6 months before discontinuation** | | | | | | |
| | No | 20.6 (20.1–21.1) | 46.5 (45.8–47.1) | 70.3 (69.7–70.9) | Reference | Reference |
| | Yes | 26.1 (25.4–26.8) | 52.7 (51.9–53.5) | 76.1 (75.3–76.7) | 1.19 (1.17–1.22) | 1.04 (1.01–1.07) |

* Adjusted for all other covariates in the table.

† In the New Zealand healthcare system, people can record up to six ethnic groups. For statistical purposes, each individual can be allocated to a single ethnic group using a prioritisation algorithm [32]. The MELAA group (Middle Eastern, Latin American, African) was included in Other.

‡ Of the 38,833 people included in this analysis, ethnicity was unknown for 1,346, NZDep13 was unknown for 28, and District Health Board was unknown for 671.

§ Record of a laboratory test in the 'blood glucose' category (includes HbA1c, fructosamine, glucose tolerance, and serum glucose tests)

another antidiabetic regimen; by 5 years the figures were 73% and 5%, respectively. Subsequent discontinuation following the first reinitiation was common (48% after 1 year). Discontinuation and reinitiation varied by age, ethnicity, and other person- and healthcare-related factors, as did the discontinuation rate, a measure of adherence volatility.

Most of the associations we observed between person- and healthcare-related factors and discontinuation, reinitiation, and adherence volatility were consistent with our previous study

**Table 3. Rate of discontinuation of metformin monotherapy and rate ratios by person- and healthcare-related factors at cohort entry.**

| Person- or healthcare-related factor | | Rate per 10 person-years (95% CI) | Crude rate ratio (95% CI) | Adjusted rate ratio* (95% CI) |
|---|---|---|---|---|
| **Age at cohort entry (years)** | | | | |
| | <25 | 4.12 (3.87–4.39) | 1.31 (1.23–1.39) | 1.20 (1.12–1.28) |
| | 25–34 | 3.98 (3.84–4.12) | 1.26 (1.22–1.31) | 1.17 (1.13–1.21) |
| | 35–44 | 3.15 (3.09–3.21) | Reference | Reference |
| | 45–54 | 2.35 (2.29–2.42) | 0.75 (0.73–0.77) | 0.81 (0.79–0.83) |
| | 55–64 | 1.65 (1.60–1.69) | 0.52 (0.51–0.54) | 0.64 (0.62–0.66) |
| | 65–74 | 1.24 (1.21–1.28) | 0.40 (0.38–0.41) | 0.54 (0.52–0.56) |
| | ≥75 | 1.18 (1.14–1.23) | 0.38 (0.36–0.39) | 0.58 (0.55–0.60) |
| **Gender** | | | | |
| | Female | 1.99 (1.97–2.02) | Reference | Reference |
| | Male | 1.96 (1.92–2.00) | 0.98 (0.96–1.00) | 0.99 (0.97–1.01) |
| **Prioritised ethnicity** | | | | |
| | Māori | 2.76 (2.69–2.83) | 1.90 (1.85–1.95) | 1.54 (1.50–1.58) |
| | Pacific | 3.17 (3.10–3.25) | 2.18 (2.13–2.24) | 1.64 (1.59–1.69) |
| | European | 1.45 (1.43–1.47) | Reference | Reference |
| | Asian (non-Indian | 1.95 (1.88–2.02) | 1.34 (1.29–1.39) | 1.11 (1.07–1.15) |
| | Indian | 2.33 (2.25–2.41) | 1.60 (1.55–1.66) | 1.23 (1.18–1.27) |
| | Other | 2.14 (1.98–2.30) | 1.47 (1.37–1.58) | 1.18 (1.10–1.27) |
| **Socioeconomic deprivation (NZDep13) at cohort entry** | | | | |
| | Quintile 1 | 1.67 (1.62–1.71) | Reference | Reference |
| | Quintile 2 | 1.74 (1.68–1.81) | 1.05 (1.01–1.09) | 1.00 (0.96–1.03) |
| | Quintile 3 | 1.74 (1.68–1.81) | 1.05 (1.01–1.09) | 1.00 (0.96–1.04) |
| | Quintile 4 | 1.88 (1.82–1.95) | 1.13 (1.09–1.17) | 1.02 (0.99–1.06) |
| | Quintile 5 | 2.40 (2.32–2.48) | 1.44 (1.39–1.49) | 1.07 (1.03–1.11) |
| **District Health Board at cohort entry** | | | | |
| | Auckland | 2.26 (2.20–2.32) | Reference | Reference |
| | Bay of Plenty | 1.80 (1.71–1.90) | 0.80 (0.76–0.84) | 0.90 (0.86–0.95) |
| | Canterbury | 1.68 (1.61–1.75) | 0.74 (0.71–0.78) | 0.92 (0.88–0.96) |
| | Capital and Coast | 1.86 (1.76–1.95) | 0.82 (0.78–0.86) | 0.86 (0.82–0.91) |
| | Counties Manukau | 2.46 (2.38–2.54) | 1.09 (1.05–1.12) | 1.00 (0.97–1.04) |
| | Hawkes Bay | 1.87 (1.77–1.97) | 0.83 (0.78–0.87) | 0.90 (0.86–0.95) |
| | Hutt | 1.85 (1.75–1.97) | 0.82 (0.77–0.87) | 0.86 (0.81–0.91) |
| | Lakes | 2.01 (1.88–2.16) | 0.89 (0.83–0.95) | 0.92 (0.86–0.99) |
| | MidCentral | 1.75 (1.65–1.85) | 0.77 (0.73–0.82) | 0.92 (0.87–0.98) |
| | Nelson Marlborough | 1.59 (1.48–1.71) | 0.70 (0.65–0.76) | 0.88 (0.82–0.95) |
| | Northland | 2.01 (1.91–2.12) | 0.89 (0.85–0.94) | 0.95 (0.90–1.00) |
| | South Canterbury | 1.41 (1.26–1.56) | 0.62 (0.56–0.69) | 0.84 (0.76–0.94) |
| | Southern | 1.44 (1.37–1.52) | 0.64 (0.61–0.67) | 0.85 (0.81–0.89) |
| | Tairawhiti | 2.27 (2.08–2.47) | 1.00 (0.92–1.09) | 0.95 (0.87–1.03) |
| | Taranaki | 1.49 (1.39–1.60) | 0.66 (0.61–0.71) | 0.83 (0.78–0.89) |
| | Waikato | 1.95 (1.86–2.03) | 0.86 (0.82–0.90) | 0.97 (0.93–1.01) |
| | Wairarapa | 1.70 (1.53–1.89) | 0.75 (0.68–0.84) | 0.94 (0.85–1.04) |
| | Waitemata | 2.05 (1.97–2.12) | 0.91 (0.87–0.94) | 1.01 (0.97–1.04) |
| | West Coast | 1.53 (1.33–1.76) | 0.68 (0.59–0.78) | 0.89 (0.77–1.02) |
| | Whanganui | 1.86 (1.72–2.01) | 0.82 (0.76–0.89) | 0.91 (0.84–0.98) |
| **Charlson Comorbidity Index** | | | | |
| | 0 | 2.04 (2.02–2.06) | Reference | Reference |

*(Continued)*

**Table 3.** (Continued)

| Person- or healthcare-related factor | | Rate per 10 person-years (95% CI) | Crude rate ratio (95% CI) | Adjusted rate ratio* (95% CI) |
|---|---|---|---|---|
| | 1 | 1.59 (1.53–1.64) | 0.78 (0.75–0.80) | 1.00 (0.96–1.04) |
| | 2 | 1.38 (1.30–1.47) | 0.67 (0.63–0.72) | 0.97 (0.91–1.04) |
| | ≥3 | 1.34 (1.21–1.47) | 0.65 (0.59–0.72) | 1.02 (0.92–1.12) |
| **History of cardiovascular disease at cohort entry** | | | | |
| | No | 2.17 (2.15–2.19) | Reference | Reference |
| | Yes | 1.37 (1.33–1.40) | 0.63 (0.61–0.65) | 0.85 (0.82–0.87) |
| **Cancer registration in the year before cohort entry** | | | | |
| | No | 1.98 (1.96–2.00) | Reference | Reference |
| | Yes | 1.50 (1.32–1.71) | 0.76 (0.67–0.87) | 0.94 (0.82–1.07) |
| **Number of hospitalisations in the year before cohort entry** | | | | |
| | 0 | 1.98 (1.96–2.01) | Reference | Reference |
| | 1 | 1.99 (1.94–2.05) | 1.00 (0.98–1.03) | 1.09 (1.06–1.12) |
| | 2–4 | 1.81 (1.74–1.89) | 0.91 (0.88–0.95) | 1.10 (1.06–1.15) |
| | 5–9 | 1.80 (1.59–2.03) | 0.91 (0.80–1.02) | 1.26 (1.12–1.43) |
| | ≥10 | 1.58 (1.11–2.25) | 0.79 (0.56–1.13) | 1.11 (0.78–1.58) |
| **Depression in the 6 months before cohort entry** | | | | |
| | No | 2.00 (1.98–2.02) | Reference | Reference |
| | Yes | 1.69 (1.63–1.75) | 0.84 (0.82–0.87) | 1.01 (0.98–1.05) |
| **Number of non-diabetic medications in the 6 months before cohort entry** | | | | |
| | 0–1 | 2.87 (2.81–2.94) | Reference | Reference |
| | 2–3 | 2.40 (2.33–2.47) | 0.83 (0.81–0.86) | 0.88 (0.86–0.91) |
| | 4–5 | 2.05 (1.99–2.11) | 0.71 (0.69–0.74) | 0.81 (0.79–0.84) |
| | 6–7 | 1.71 (1.66–1.77) | 0.60 (0.58–0.62) | 0.72 (0.69–0.74) |
| | 8–9 | 1.54 (1.48–1.60) | 0.54 (0.52–0.56) | 0.68 (0.65–0.70) |
| | 10–19 | 1.40 (1.35–1.45) | 0.49 (0.47–0.51) | 0.64 (0.62–0.67) |
| | ≥20 | 1.18 (1.07–1.32) | 0.41 (0.37–0.46) | 0.57 (0.51–0.64) |
| **Glucose test in the 6 months before cohort entry†** | | | | |
| | No | 2.42 (2.37–2.47) | Reference | Reference |
| | Yes | 1.90 (1.86–1.95) | 0.79 (0.77–0.81) | 0.90 (0.88–0.93) |
| **Urinary albumin/creatinine ratio test in the 6 months before cohort entry** | | | | |
| | No | 2.01 (1.99–2.03) | Reference | Reference |
| | Yes | 1.92 (1.89–1.96) | 0.96 (0.94–0.98) | 0.98 (0.96–1.00) |

* Adjusted for all other covariates in the table.

† In the New Zealand healthcare system, people can record up to six ethnic groups. For statistical purposes, each individual can be allocated to a single ethnic group using a prioritisation algorithm [32]. The MELAA group (Middle Eastern, Latin American, African) was included in Other.

‡ Of the 85,066 people included in this analysis, ethnicity was unknown for 3,065, NZDep13 was unknown for 4, and District Health Board was unknown for 9.

§ Record of a laboratory test in the 'blood glucose' category (includes HbA1c, fructosamine, glucose tolerance, and serum glucose tests).

and findings of studies conducted in other settings [10, 33, 34]. Age and ethnicity are two particularly important factors to consider. Better adherence and persistence in older people with T2DM is a common finding in the literature [17, 33–36]. In our study, older age was associated with a lower risk of discontinuation (both initially and after the first reinitiation) compared with younger age. However, older age was also associated with a reduced likelihood of reinitiation after the first discontinuation. These findings may reflect differing reasons for discontinuing metformin in older versus younger cohort members and suggest a need for different approaches to facilitate adherence in younger people with T2DM. This is important given the

high levels of discontinuation in the younger age groups (more than half of cohort members aged < 35 years discontinued metformin monotherapy within 1 year of initiation), the decreasing age of onset of T2DM in New Zealand [5], and the association between younger age at onset and poorer outcomes [6].

Ethnicity was also strongly associated with metformin monotherapy discontinuation and reinitiation. Ethnic groups disproportionately affected by T2DM (Māori, Pacific peoples, Indian) were more likely to discontinue (both initially and after the first reinitiation) than other ethnic groups. However, they were also more likely to reinitiate following a discontinuation, which reflects a cyclical pattern of use. These findings are consistent with earlier work based on the national routine data collections that suggested Māori and Pacific peoples receive fewer prescriptions for oral hypoglycaemics as a medication class after adjusting for T2DM disease burden, and are less likely to continue with these medications when they do receive them [37, 38]. While financial costs have been shown to be a particular barrier for Māori and Pacific peoples in accessing health services and pharmaceuticals [3, 39], this is unlikely to fully explain the greater adherence volatility we observed [39]. A systematic review of Māori experiences with the health system, and the contribution of these experiences to health inequities, has highlighted the patient-clinician relationship and communication as crucial contributors and has pointed to a range of potential strategies at the clinical and structural level that could be used to reduce these inequities [40]. Given the increasing disparity in the prevalence of T2DM in Māori and Pacific populations in particular, our findings are extremely concerning and point to an urgent need to implement and evaluate the strategies recommended by the Waitangi Tribunal [41] to reorient health service delivery, particularly in primary care, to achieve health equity. This is a point of global relevance; the disproportionate impact of T2DM on indigenous communities and the contribution of health service delivery to inequitable health outcomes is not limited to New Zealand [1, 40].

A key strength of this study is that it provides a comprehensive, national-level view of the patterns of discontinuation and reinitiation of the recommended first-line pharmacological therapy for T2DM. The study cohort was derived from a validated data source, the VDR [21, 23, 24], which excludes women with gestational diabetes and polycystic ovary syndrome (who might have received treatment with metformin). This, combined with the additional steps we took to exclude people with type 1 diabetes mellitus, increases the certainty that cohort members were dispensed metformin for T2DM. We are also likely to have identified nearly all metformin dispensings to cohort members. The cost of the vast majority of medications commonly used outside of hospitals, such as metformin, is subsidised by the state for all New Zealand residents [42]. To receive payment for this subsidy, community pharmacies must claim for it by submitting details of a dispensing, and of the person to whom the medicine was dispensed, to Pharms. This means that Pharms captures virtually all of the dispensings by community pharmacies in New Zealand. While we will not have captured metformin dispensings during hospital admissions or any which occurred during extended overseas stays, this is likely to have had a negligible impact on our findings; for instance, among cohort members who were hospitalised during the study period, 95% of hospitalisations were ≤ 14 days in duration and only 1.5% lasted > 30 days.

As with all medication adherence research that is based on electronic health records, it was impossible to determine whether, and when, cohort members took the medication they were dispensed. While this might have resulted in some imprecision in the dates ascribed to the start and end of gaps in metformin monotherapy use, other research has suggested good concordance between prescription databases and physical pill counts [43], suggesting that the impact of this on the validity our findings is likely to be small. A further limitation is that we did not have access to detailed clinical data, including the results of HbA1c tests, so we were

unable to explore the impact of discontinuations on glycaemic control. Additionally, although we had some data on health events and health status, there were still gaps meaning that residual confounding by comorbidity may have still been present.

While this study only assesses dispensing patterns up to 2014, we are confident that prescribing practices are unlikely to have materially changed since then. The guidelines promulgated to prescribers for initiating pharmacological therapy for T2DM in New Zealand have not changed since the study [5]. Another study using slightly more recent dispensing data indicated an increasing trend towards people with T2DM initiating therapy with metformin in accordance with these guidelines [9].

In conclusion, this national cohort study of new users of metformin monotherapy has shown that some groups spend substantial periods in a state of discontinuation, or cycle rapidly between discontinuation and reinitiation states. These findings highlight the importance of interventions at the level of individual patients and patient/healthcare provider consultations, as well as the need to address broader structural factors which act as barriers to long-term adherence and contribute to inequities in access to medicines and their optimal use.

## Supporting information

**S1 Fig. Derivation of the study cohort.** The numbers represent the people in the cohort after the previous criteria have been applied.
(TIF)

**S2 Fig. Cumulative proportions who discontinued metformin monotherapy at least once, who changed–another antidiabetic regimen, and who died after reinitiation of metformin monotherapy.**
(TIF)

**S1 Table. Comparison of cumulative proportions who discontinued metformin monotherapy after first reinitiation by person- and healthcare-related factors.**
(DOCX)

## Acknowledgments

We would like to thank Analytical Services at the Ministry of Health for supplying the data, the Medical Council of New Zealand for providing information on medical practitioners' vocational scope, and the advisory group for this project for their helpful comments (Jason Arnold, Deborah Connor, Dr Sisira Jayathissa, Dr Jesse Kokaua, Dr Jeremy Krebs, Dr Andrew Sporle and Dr Michael Tatley).

## Author Contributions

**Conceptualization:** Simon Horsburgh, Katrina Sharples, David Barson, Jiaxu Zeng, Lianne Parkin.

**Data curation:** David Barson.

**Formal analysis:** Katrina Sharples.

**Funding acquisition:** Simon Horsburgh, Katrina Sharples, David Barson, Jiaxu Zeng, Lianne Parkin.

**Methodology:** Simon Horsburgh, Katrina Sharples, David Barson, Jiaxu Zeng, Lianne Parkin.

**Project administration:** Lianne Parkin.

**Visualization:** Katrina Sharples.

**Writing – original draft:** Simon Horsburgh.

**Writing – review & editing:** Simon Horsburgh, Katrina Sharples, David Barson, Jiaxu Zeng, Lianne Parkin.

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
