## [Decision Letter · Decision Letter 0]

11 Feb 2021

PONE-D-20-37836

Patterns of metformin monotherapy discontinuation and reinitiation in people with type 2 diabetes mellitus in New Zealand.

PLOS ONE

Dear Dr. Horsburgh,

Thank you for submitting your manuscript to PLOS ONE. After careful consideration, we feel that it has merit but does not fully meet PLOS ONE’s publication criteria as it currently stands. Therefore, we invite you to submit a revised version of the manuscript that addresses the points raised during the review process.

We look forward to receiving your revised manuscript.

Kind regards,

Jingjing Qian

Academic Editor

PLOS ONE

Additional Editor Comments:

Please be aware of and properly address the data availability requirements of PLOS ONE.

Journal Requirements:

2. We noted in your submission details that a portion of your manuscript may have been presented or published elsewhere.

"As described in the Letter to the Editor, a summary of the methods and some of the findings from this study were presented as a short oral presentation at a Symposium, with the abstract published and freely available.  We do not consider this dual publication due to the limited and superficial coverage of the findings at the Symposium, and the published abstract only containing a narrative summary of the findings (and not reporting any specific results)."

Reviewers' comments:

Reviewer's Responses to Questions

**Comments to the Author**

1. Is the manuscript technically sound, and do the data support the conclusions?

Reviewer #1: Yes

Reviewer #2: Yes

2. Has the statistical analysis been performed appropriately and rigorously? 

Reviewer #1: Yes

Reviewer #2: Yes

3. Have the authors made all data underlying the findings in their manuscript fully available?

Reviewer #1: No

Reviewer #2: Yes

4. Is the manuscript presented in an intelligible fashion and written in standard English?

Reviewer #1: Yes

Reviewer #2: Yes

5. Review Comments to the Author

Reviewer #1: The manuscript submitted to PLOS ONE by Horsburgh and colleagues entitled "Patterns of metformin monotherapy discontinuation and reinitiation in people with type 2 diabetes mellitus in New Zealand" describes the patterns of discontinuation and reinitiation among patients newly initiating metformin using New Zealand Ministry of Health's Virtual Diabetes Register which include health and pharmaceutical dispensing data. I congratulate the authors for their thorough work in investigating this important issue, and for the well organized well written manuscript. I have only a several questions/suggestions for them:

- A flow diagram in Suppl. Figure 1 may be more informative with the number of patients affected by each excluding condition.

- (On a similar note as above,) How many patients were impacted by excluding "patients who permanently discontinued metformin <100 days of initiation?"

- More information on the medication dispensing data and the insurance system in New Zealand would be helpful to understand the validity of the study. Are all dispensations captured by the database regardless of the insurance type?

- Is the medication for oral diabetes treatment predominantly metformin in New Zealand? What are the percentages of other antidiabetics used as the initial treatment?

- The authors should explain the rational for including the presence of glucose test within 6 month as well as urinary ALB/CRE ratio test as covariates. What were they looking for and how are they interpreting their association with the outcomes?

Reviewer #2: Dear Authors, this is an interesting study evaluating prescription patterns of metformin in T2DM in New Zeland from 2006 and 2014. These findings will contribute to give important tools for the improvement of medication adherence. This study investigates a phenomenon in a timeframe of 8 years, therefore very consistent and scientifically relevant, I am only concerned that, currently, after 7 years, the prescribing patterns in New Zealand may have changed. It would therefore be of very interest to conduct a second study, using the same methodology, in a timeframe of 2014-2020.

I’ve only one concern about the manuscript, as follow:

Since the present work focuses on the evaluation of pattern of discontinuation and reinitiation to treatment, two phases of medication adherence, it is necessary, in the introduction, to better define the concept of adherence to treatment by citing the current and recognized EMERGE guidelines on Medication Adherence (available from: https://www.espacomp.eu/project/emerge-guidelines/), explaining the 3 different phases of adherence; in addition, in the methodology, it should be subsequently indicated in detail which of the three phases should be studied.

6. PLOS authors have the option to publish the peer review history of their article (what does this mean?). If published, this will include your full peer review and any attached files.

Reviewer #1: No

Reviewer #2: No

---

## [Author Response · Author response to Decision Letter 0]

4 Mar 2021

(these are formatted more pleasantly in the Responses to Reviewers document).

Responses to Reviewers for PONE-D-20-37836: Patterns of metformin monotherapy discontinuation and reinitiation in people with type 2 diabetes mellitus in New Zealand.

We would like to thank the reviewers for their thoughtful and constructive comments and suggestions on the manuscript. Please find our responses to these below.

Editorial Queries

We have reviewed the manuscript and ensured that it meets PLOS One’s style requirements.

We have addressed the other editorial queries in the accompanying cover letter as requested.

 

Reviewer #1

1. A flow diagram in Suppl. Figure 1 may be more informative with the number of patients affected by each excluding condition.

Thank you to the reviewer for making this helpful suggestion. The study flowchart has been updated to include the number of people in the study cohort at each point in the cohort's derivation.

2. (On a similar note as above,) How many patients were impacted by excluding "patients who permanently discontinued metformin <100 days of initiation?"

A total of 3,952 (or 4.4%) of the study cohort at that point in the study cohort derivation were excluded because of the ‘permanently discontinued metformin within 100 days of the first metformin dispensing’ rule.

3. More information on the medication dispensing data and the insurance system in New Zealand would be helpful to understand the validity of the study. Are all dispensations captured by the database regardless of the insurance type?

Thank you for this suggestion; we always incorrectly assume everyone is familiar with New Zealand’s health system!

New Zealand has universal health coverage, with most health care publicly funded [1]. This includes prescription medicines. The cost of most medicines used in New Zealand is subsidised, either fully or partially, by the state Pharmaceutical Management Agency (PHARMAC). Private insurance has little involvement with prescription medicines, except in the case of a very small set of high-cost medicines not subsidised fully by the state (such as certain cancer treatments). With regard to medicines commonly dispensed outside of hospitals, prescription medicines can effectively be considered to be publicly funded for all residents. To be paid for a dispensing, community pharmacies must submit claims for payment, including details about a dispensing and the patient dispensed to. This information is collected in the Pharmaceutical Collection (Pharms), which provides the dispensing data used in this study. Because the vast majority of medicines used in the community are state subsidised, including metformin, and community pharmacies are required to claim reimbursement for subsidised dispensings, Pharms provides virtually complete capture of community pharmacy dispensings to the New Zealand population. This is a substantial advantage to using the New Zealand dispensing data collection.

To briefly clarify the nature and capture of Pharms, we have added the following to Section 2.3 (Construction of medication record):

Pharms contains details of community pharmacy dispensings of medications subsidised by the state (virtually all commonly used medications in New Zealand), providing a comprehensive information source on community medication use.

We have also expanded our brief sentence on the capture of Pharms in the Discussion to now read:

We are also likely to have identified nearly all metformin dispensings to cohort members. The cost of the vast majority of medications commonly used outside of hospitals, such as metformin, is subsidised by the state for all New Zealand residents [41]. To receive payment for this subsidy, community pharmacies must claim for it by submitting details of a dispensing, and of the person to whom the medicine was dispened, to Pharms. This means that Pharms captures virtually all of the dispensings by community pharmacies in New Zealand.

Further information on Pharms and the other data sources cited in this manuscript are available in our previous manuscript based on the larger study [2] and cited in Section 2.3 (Construction of medication record).

4. Is the medication for oral diabetes treatment predominantly metformin in New Zealand? What are the percentages of other antidiabetics used as the initial treatment?

Metformin is the predominant initial treatment for type 2 diabetes mellitus in New Zealand. A recent national study looking at initial pharmacotherapy for T2DM found that, in 2015/2016, 85% of New Zealanders with T2DM commenced pharmacotherapy with metformin monotherapy. [3] Metformin with a sulfonylurea (6%) and metformin with an insulin (5%) were the other combinations to occur more than 2% of the time in the data.

We have amended the first sentence of the second paragraph in the Introduction to now read:

Metformin monotherapy is the first-line pharmacological treatment for T2DM in New Zealand [4,5] and accounts for approximately 85% of the initial pharmacological agents prescribed for T2DM [3]

5. The authors should explain the rational for including the presence of glucose test within 6 month as well as urinary ALB/CRE ratio test as covariates. What were they looking for and how are they interpreting their association with the outcomes?

We included these two covariates for two main reasons. Firstly, we were interested to see what proportion of people with T2DM initiating metformin monotherapy for the first time actually had a recent glucose test, given that the guidelines for diagnosis and management of T2DM in New Zealand explicitly refer to HbA1c level thresholds.[4] Secondly, we saw these as proxies for monitoring of T2DM by a health professional, with the hypothesis that increased monitoring by a health professional would improve adherence and reinitiation (this has been found elsewhere e.g. [6,7]). These tests are recommended as part of clinical monitoring of people with T2DM in New Zealand, and so we felt they provided a relevant proxy for clinical monitoring and engagement.

This manuscript's findings showed a decreased risk of first discontinuation, increased risk of subsequent reinitiation, and lower rates of discontinuation when these tests were performed in the six months before cohort entry or discontinuation of metformin monotherapy. These associations are consistent with what one would expect based on the studies cited above and the hypothesis that more intensive clinical monitoring improves adherence to and reinitiation of metformin monotherapy. 

We have added the following to the bottom of Section 2.5 (Covariates) to clarify why the covariates were selected:

These covariates were chosen because they represented important population groups, were factors identified in the literature as potentially influencing metformin adherence, or were tests recommended in New Zealand as part of clinical monitoring for people with T2DM.

 

Reviewer Two

1. I am only concerned that, currently, after 7 years, the prescribing patterns in New Zealand may have changed. It would therefore be of very interest to conduct a second study, using the same methodology, in a timeframe of 2014-2020.

We agree that it would be good to assess patterns of metformin monotherapy using more recent data not just to examine changes in prescribing patterns but also to monitor whether inequities between groups have changed. For the current study, we were limited in the recency of dispensing data by the lag in the availability of complete datasets (particularly hospitalisations and mortality datasets) that we relied upon for other parts of the wider project not presented in this paper. We hope that this study will serve as an important baseline for future research looking at longer time trends.

We would note that guidelines for initiating pharmacological therapy for T2DM have not changed in New Zealand since the study.

2. Since the present work focuses on the evaluation of pattern of discontinuation and reinitiation to treatment, two phases of medication adherence, it is necessary, in the introduction, to better define the concept of adherence to treatment by citing the current and recognized EMERGE guidelines on Medication Adherence (available from: https://www.espacomp.eu/project/emerge-guidelines/), explaining the 3 different phases of adherence; in addition, in the methodology, it should be subsequently indicated in detail which of the three phases should be studied.

We are familiar with the EMERGE Guidelines [8], which provide an excellent framework for reporting adherence studies, and have followed its guidance in this manuscript where applicable. However, our manuscript reports on a study of dynamic changes in adherence using the ‘refill-gap method’ analytical approach described by Grégoire and Moisan [9] which, we feel, does not elegantly fit within Vrijens et al.’s taxonomy of medication adherence as published (e.g. [10,11]). Since the EMERGE Guidelines mandate using Vrijens et al.’s taxonomy, we have chosen not to comply with the Guidelines fully. We hope the following discussion will explain our decision.

Vrijens et al.’s initiation, implementation and persistence framework provides a useful overarching conceptual guide and consistent vocabulary for adherence research. However, we feel it becomes unwieldy in the context of considering adherence as a dynamic process, which is where we have explicitly situated it here. If persistence is defined as the time from the first prescription to discontinuation (as per Vrijens et al.), then where does subsequent reinitiation fit in this taxonomy (it is not listed as a phase in the published papers describing the taxonomy)? Does reinitiation constitute the initiation of a new adherence ‘episode’? Or, because the discontinuation was not permanent, does it not count as a true discontinuation, and therefore the person never actually leaves the implementation phase? Researchers such as Jensen et al. [12,13] have taken the former approach whilst the EMERGE Guidelines seem to imply the latter. Others (e.g. [14]) have used the EMERGE Guidelines whilst tacitly noting the limitations of Vrijens et al.’s taxonomy for adherence research which looks at medication adherence as a dynamic process. We would note that Vrijens et al.’s taxonomy is also being stretched by further developments in the field, such as the use of trajectory models to classify patterns of adherence behaviour.

As the above discussion suggests, we are not clear how our study would fit into the three phases of Vrijens et al.’s taxonomy. Instead, we have followed the ‘refill-gap method’ analytical approach described by Grégoire and Moisan [9]. Thus, we are not sure how describing the taxonomy nor using it to frame the reporting of our findings would add to the manuscript or help the reader understand what we did and its implications. That said, we do note that our use of a time to event analysis for calculating associations with the first discontinuation follows best practice suggested by Vrijens et al. [10,11] and Halpern et al. [15]. We have added the following (with citations) to the beginning of Section 2.6 (Statistical analyses) to make this approach explicit:

We used the refill-gap method as a framework for our analyses [23], along with time-to-event analyses as recommended by Vrijens et al. [24,25]. 

In keeping with the EMERGE Guidelines, and the helpful conceptual clarity that Vrijens et al.’s taxonomy provides, we have substituted the term ‘persistence’ and its variations with ‘continued’, ‘being dispensed’, ‘adherence’, ‘discontinuation and reinitiation’, ‘medication use’ or ‘long-term adherence’ throughout the manuscript as appropriate to avoid ambiguity given that it is unclear whether what we describe would be considered persistence according with the EMERGE Guidelines and Vrijens et al.’s taxonomy. 

 

References

1. Gauld R. The New Zealand Health Care System. In: Tikkanen R, Osborn R, Mossialos E, Djordjevic A, Wharton G, editors. International Profiles of Health Care Systems. New York; 2020. pp. 149–158. 

2. Horsburgh S, Barson D, Zeng J, Sharples K, Parkin L. Adherence to metformin monotherapy in people with type 2 diabetes mellitus in New Zealand. Diabetes Res Clin Pract. 2019;158: 107902. doi:10.1016/j.diabres.2019.107902

3. Murray P, Norris H, Metcalfe S, Betty B, Young V, Locke B. Dispensing patterns for antidiabetic agents in New Zealand: Are the guidelines being followed? N Z Med J. 2017;130: 12–18. 

4. New Zealand Guidelines Group. Guidance on the management of type 2 diabetes. Wellington: New Zealand Guidelines Group; 2011. 

5. bpacNZ. Managing patients with type 2 diabetes: from lifestyle to insulin. Best Pract J. 2015; 32–42. 

6. van Wijk BL, Avorn J, Solomon DH, Klungel OH, Heerdink ER, de Boer A, et al. Rates and determinants of reinitiating antihypertensive therapy after prolonged stoppage: a population-based study. J Hypertens. 2007;25: 689–697. doi:10.1097/HJH.0b013e3280148a58

7. Brookhart MA, Patrick AR, Schneeweiss S, Avorn J, Dormuth C, Shrank W, et al. Physician follow-up and provider continuity are associated with long-term medication adherence: a study of the dynamics of statin use. Arch Intern Med. 2007;167: 847–52. doi:10.1001/archinte.167.8.847

8. De Geest S, Zullig LL, Dunbar-Jacob J, Helmy R, Hughes DA, Wilson IB, et al. ESPACOMP Medication Adherence Reporting Guideline (EMERGE). Ann Intern Med. 2018;169: 30. doi:10.7326/M18-0543

9. Grégoire J-P, Moisan J. Assessment of adherence to drug treatment in database research. In: Elseviers M, Wettermark B, Almarsdóttir AB, Andersen M, Benko R, Bennie M, et al., editors. Drug Utilization Research. Chichester, UK: John Wiley & Sons, Ltd; 2016. pp. 369–380. doi:10.1002/9781118949740.ch36

10. Vrijens B, De Geest S, Hughes DA, Przemyslaw K, Demonceau J, Ruppar T, et al. A new taxonomy for describing and defining adherence to medications. Br J Clin Pharmacol. 2012;73: 691–705. doi:10.1111/j.1365-2125.2012.04167.x

11. Vrijens B, Elseviers M, Vrijens B, Grégoire J-P, Moisan J. An introduction to adherence research. Drug Utilization Research. 2016. pp. 355–360. doi:https://doi.org/10.1002/9781118949740.ch34

12. Jensen ML, Jørgensen ME, Hansen EH, Aagaard L, Carstensen B. A multistate model and an algorithm for measuring long-term adherence to medication: a case of diabetes mellitus type 2. Value Health. 2014;17: 266–74. doi:10.1016/j.jval.2013.11.014

13. Jensen ML, Jørgensen ME, Hansen EH, Aagaard L, Carstensen B. Long-term patterns of adherence to medication therapy among patients with type 2 diabetes mellitus in Denmark: the importance of initiation. PLoS One. 2017;12: e0179546. doi:10.1371/journal.pone.0179546

14. Alfian SD, Denig P, Coelho A, Hak E. Pharmacy-based predictors of non-adherence, non-persistence and reinitiation of antihypertensive drugs among patients on oral diabetes drugs in the Netherlands. PLoS One. 2019;14: e0225390. doi:10.1371/journal.pone.0225390

15. Halpern MT, Khan ZM, Schmier JK, Burnier M, Caro JJ, Cramer J, et al. Recommendations for evaluating compliance and persistence with hypertension therapy using retrospective data. Hypertension. 2006;47: 1039–1048. doi:10.1161/01.HYP.0000222373.59104.3d

---

## [Decision Letter · Decision Letter 1]

29 Mar 2021

PONE-D-20-37836R1

Patterns of metformin monotherapy discontinuation and reinitiation in people with type 2 diabetes mellitus in New Zealand.

PLOS ONE

Dear Dr. Horsburgh,

Thank you for submitting your manuscript to PLOS ONE. After careful consideration, we feel that it has merit but does not fully meet PLOS ONE’s publication criteria as it currently stands. Therefore, we invite you to submit a revised version of the manuscript that addresses the points raised during the review process.

We look forward to receiving your revised manuscript.

Kind regards,

Jingjing Qian

Academic Editor

PLOS ONE

Journal Requirements:

Additional Editor Comments (if provided):

Thanks for submitting the revision. I feel the authors fully addressed reviewer #1's comments in the revised manuscript. Although they provided detailed, appropriate explanation to reviewer #2's comments, they did not fully transfer their responses in the revised manuscript. Please 1) acknowledge the limitation of older data used in this study and justify how the current study may inform current prescribing patterns in New Zealand; and 2) briefly discuss the rationale of deviation from the EMERGE guidelines on Medication Adherence and the use of discontinuation and reinitiation to treatment in the current study, which can be incorporated in either Introduction or Methods section of the manuscript. Thanks.

Reviewers' comments:

Reviewer's Responses to Questions

**Comments to the Author**

1. If the authors have adequately addressed your comments raised in a previous round of review and you feel that this manuscript is now acceptable for publication, you may indicate that here to bypass the “Comments to the Author” section, enter your conflict of interest statement in the “Confidential to Editor” section, and submit your "Accept" recommendation.

Reviewer #1: All comments have been addressed

2. Is the manuscript technically sound, and do the data support the conclusions?

Reviewer #1: Yes

3. Has the statistical analysis been performed appropriately and rigorously? 

Reviewer #1: Yes

4. Have the authors made all data underlying the findings in their manuscript fully available?

Reviewer #1: No

5. Is the manuscript presented in an intelligible fashion and written in standard English?

Reviewer #1: Yes

6. Review Comments to the Author

Reviewer #1: All of my comments have been fully addressed.

I commend the authors for this important and well conducted study.

7. PLOS authors have the option to publish the peer review history of their article (what does this mean?). If published, this will include your full peer review and any attached files.

Reviewer #1: No

---

## [Author Response · Author response to Decision Letter 1]

30 Mar 2021

Please see cover letter for our responses. They are also copied below.

The revised manuscript includes amendments to address the suggested changes. With regard to point one, we have added the following to the Discussion section:

While this study only assesses dispensing patterns up to 2014, we are confident that prescribing practices are unlikely to have materially changed since then. The guidelines promulgated to prescribers for initiating pharmacological therapy for T2DM in New Zealand have not changed since the study [5]. Another study using slightly more recent dispensing data indicated an increasing trend towards people with T2DM initiating therapy with metformin in accordance with these guidelines [9]. 

Regarding point 2, we have added the following to the beginning of the Methods section:

Where feasible, we have conformed to the EMERGE Guidelines for reporting medication adherence studies [18]. However, we do not use the taxonomy described by Vrijens et al. [19] and mandated by the EMERGE Guidelines because we feel that it is conceptually ambiguous when considering adherence as a dynamic process as we do here. In particular, the taxonomy does not address reinitiation of therapy after discontinuation. Instead, we have used the refill-gap method as a more appropriate framework for examining the dynamics of medication adherence [20].

We have also reviewed our reference list for completeness and correctness, and have added another reference cited in the additions above. As far as we are aware, none of the references cited has been retracted. We have also reviewed the manuscript again for spelling and grammatical errors, and have fixed some that we embarrassingly missed earlier.

---

## [Editor Report · Decision Letter 2]

5 Apr 2021

Patterns of metformin monotherapy discontinuation and reinitiation in people with type 2 diabetes mellitus in New Zealand.

PONE-D-20-37836R2

Dear Dr. Horsburgh,

We’re pleased to inform you that your manuscript has been judged scientifically suitable for publication and will be formally accepted for publication once it meets all outstanding technical requirements.

Kind regards,

Jingjing Qian

Academic Editor

PLOS ONE

Additional Editor Comments (optional):

Thanks for properly addressing all suggested changes in the revised manuscript.
---

## [Editor Report · Acceptance letter]

8 Apr 2021

PONE-D-20-37836R2 

Patterns of metformin monotherapy discontinuation and reinitiation in people with type 2 diabetes mellitus in New Zealand. 

Dear Dr. Horsburgh:

I'm pleased to inform you that your manuscript has been deemed suitable for publication in PLOS ONE. Congratulations! Your manuscript is now with our production department. 

Kind regards, 

on behalf of

Dr. Jingjing Qian 

Academic Editor

PLOS ONE